# Forging a Masterpiece from Any Face: A Universal Framework for Face Stylization

## Abstract

The canonical challenge in face stylization lies in disentangling high-level semantic content, such as identity, from low-level stylistic attributes. Prevailing methods, including recent diffusion-based models, often fail to achieve a robust separation, resulting in an undesirable trade-off between style fidelity and content preservation. To address these challenges, we introduce **StyleFace**, a novel framework that treats face stylization as a targeted statistical transfer within a disentangled feature space. Our approach is a cohesive pipeline that begins with a disentangled attention module, which orthogonally projects content and style information into separate, controllable embeddings. This separation is critical, enabling our method's core: a statistical style injection layer that manipulates feature distributions to preserve identity while implanting style. To guide this transfer and ensure global coherence, the entire process is optimized using a perceptually-aligned adversarial objective that operates not on raw pixels, but on the high-level feature manifold of a Vision Transformer (ViT), enforcing perceptual and stylistic consistency. This synergistic design allows StyleFace to achieve an unprecedented balance between identity preservation and style fidelity, with comprehensive experiments demonstrating that our model consistently outperforms state-of-the-art methods.

## 1 Introduction

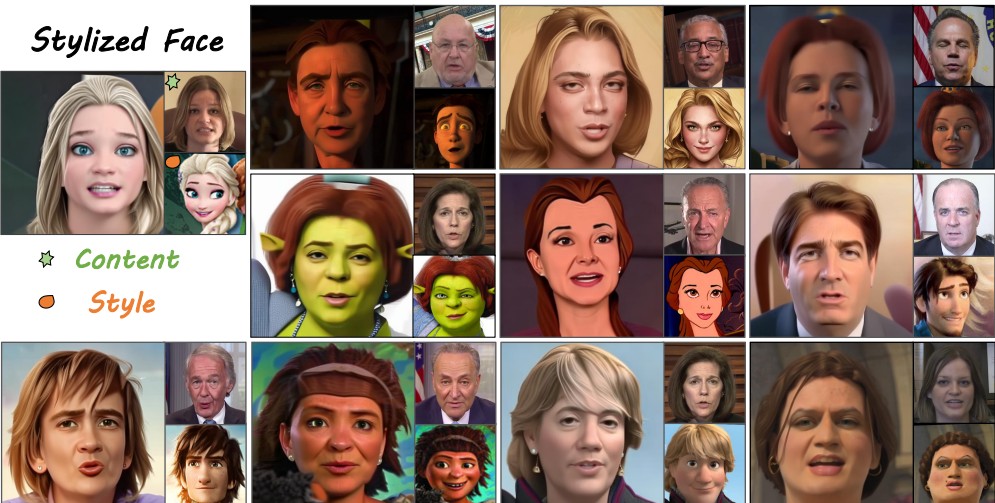

Image stylization, the transformation of images into artistic styles while preserving content characteristics Gatys et al. (2016); Kwon & Ye (2022a); Deng et al. (2022), has emerged as a significant research area in computer vision. Recent advances in diffusion models have revolutionized image generation and manipulation Ho et al. (2020); Nichol & Dhariwal (2021); Kim et al. (2022), with promising applications for style transfer Jeong et al. (2024); Wang et al. (2023); Chung et al. (2024). While general image stylization has progressed substantially Zhang et al. (2023); Hong et al. (2023); Zhang et al. (2022c); An et al. (2021); Wu et al. (2021), facial stylization introduces unique challenges due to human perceptual sensitivity to facial features and the critical need to balance artistic

transformation with identity preservation Khowaja et al. (2024); Yi et al. (2020); Shi et al. (2019). This challenging task requires specialized approaches that can effectively disentangle and recombine identity and style information while maintaining perceptual fidelity and semantic coherence across diverse artistic domains Cohen et al. (2025).

Existing approaches to face stylization encompass diverse technical paradigms, each with distinct capabilities and limitations. GAN-based methods leverage the latent space of pre-trained generators to create stylized outputs, yet they often struggle with one-shot style transfer and exhibit limited style diversity across different artistic domains Zhang et al. (2023); Yang et al. (2022); Palsson et al. (2018); Liu et al. (2019). Meanwhile, optimization-based techniques iteratively align content and style features to achieve transformation, but frequently suffer from significant identity distortion and computational inefficiency that limits their practical applications Gatys et al. (2016); Kwon & Ye (2022a); Zhang et al. (2022c); An et al. (2021); Yang et al. (2023). More recently, diffusion-based approaches have shown promise by leveraging their strong generative capabilities, yet they continue to face fundamental challenges in simultaneously maintaining facial identity while effectively transferring stylistic elements Chung et al. (2024); Jeong et al. (2024); Kwon & Ye (2022b). These limitations highlight the need for more balanced and effective approaches to facial stylization.

To address this trade-off, we propose a new conceptual framework for face stylization. We posit that the fundamental challenge of disentangling identity from style can be effectively resolved by operating on the statistical properties of feature representations within a diffusion model. Our central observation is that first-order statistics (*i.e.*, channel-wise mean) of deep features primarily encode an image's structural and identity information, while second-order statistics (*i.e.*, standard deviation) capture its style. By precisely controlling these two components, we can achieve a robust separation and recombination of content and style. Based on this principle, we introduce **StyleFace**, a framework that operationalizes targeted statistical transfer in a cohesive generative pipeline. The process begins with a novel disentangled attention mechanism that projects content and style information into separate, controllable embeddings. This critical separation enables the core of our method: a targeted statistical manipulation where we preserve the feature mean to maintain the subject's identity while aligning the feature standard deviation with that of the style reference. To ensure the final output is both semantically coherent and stylistically faithful, the entire generative process is guided by a perceptual adversarial objective. This objective operates not on raw pixels but on the high-level feature manifold of a Vision Transformer (ViT), enforcing global consistency.

Our contributions can be summarized as follows:

- We propose StyleFace, a novel diffusion-based framework that effectively resolves the fundamental tension between identity preservation and style transfer in face stylization through a principled controllable design.

- We develop a style controller that integrates facial features within diffusion models' self-attention layers to preserve initial structural identity, complemented by an identity alignment strategy that reinforces identity consistency during the generation process.

- We introduce a statistical alignment technique that explicitly disentangles content structure from style attributes and semantic-guided adversarial optimization that ensures coherent style transfer at multiple perceptual levels, enabling high-quality face stylization.

## 2 RELATED WORK

### 2.1 IMAGE STYLE TRANSFER

Image style transfer has been an active research area in computer vision. Early approaches like Gatys et al. Gatys et al. (2016) pioneered neural style transfer by using CNN features to separate and recombine content and style. Since then, numerous methods have been proposed to improve style transfer quality and efficiency Johnson et al. (2016); Huang & Belongie (2017). Transformer-based methods like StyTr2 Deng et al. (2022) leverage self-attention mechanisms for better feature transformation. CLIPStyler Kwon & Ye (2022a) enables text-guided style transfer by utilizing CLIP's multimodal representations. With the advent of diffusion models Nichol & Dhariwal (2021), several works have explored their potential for style transfer. Wang et al. Wang et al. (2023) proposed StyleDiffusion for controllable disentangled style transfer. Zhang et al. Zhang et al. (2023) developed

an inversion-based approach for more precise style control. Domain-aware methods like Zhang et al. (2022c) incorporate contrastive learning to better capture domain-specific style features.

Recent works have also focused on improving artistic quality while maintaining content fidelity. AesPA-Net Hong et al. (2023) introduces aesthetic pattern awareness for more visually pleasing results. ArtFlow An et al. (2021) employs reversible neural flows to achieve unbiased style transfer. Yang et al. Yang et al. (2023) proposed a zero-shot contrastive loss for text-guided diffusion style transfer, demonstrating the potential of diffusion models in this domain.

## 2.2 FACIAL IMAGE STYLIZATION

Recognizing the unique challenges of facial stylization, researchers have developed specialized approaches for face images Khowaja et al. (2024); Ji et al. (2024); Li et al. (2025). Early attempts adapted general style transfer methods with face-specific constraints Selim et al. (2016). More recent works have incorporated facial landmark detection Zhang et al. (2020) and identity preservation mechanisms Xu et al. (2024a). StyleGAN-based approaches Karras et al. (2019; 2020b) and unsupervised image-to-image translation methods have shown promise in facial stylization tasks like selfie-to-anime conversion. However, these methods typically require task-specific training to achieve satisfactory results. While many existing approaches excel at single-style transfer, they often struggle with multi-style fusion and customized style combinations. Methods like DreamBooth Ruiz et al. (2022) and Textual Inversion Gal et al. (2022) face similar challenges when attempting to combine multiple artistic styles like cartoon, anime, and arcane aesthetics. A key limitation across these approaches is their difficulty in preserving facial identity features, poses, and characteristics during style transfer. The misalignment between source and target domain facial features frequently results in artifacts that compromise the quality of the stylized output.

Our work advances the field by introducing a novel approach that achieves a principled balance between identity preservation and style transfer. Unlike previous methods that struggle with one-shot style transfer or suffer from identity distortion, our approach leverages the generative capabilities of latent diffusion models while introducing specialized mechanisms for disentangling and recombining identity and style features. By simultaneously operating on self-attention mechanisms, channel statistics, and semantic embeddings, our method ensures high-level stylistic coherence while maintaining facial identity, addressing key limitations in existing approaches.

## 3 METHOD

In this paper, we propose StyleFace for face stylization that utilizes the benefits of diffusion models in generative ability. Our method is designed to preserve the identity of the subject while transferring the style of a reference image. The framework of our method is presented in Figure 1.

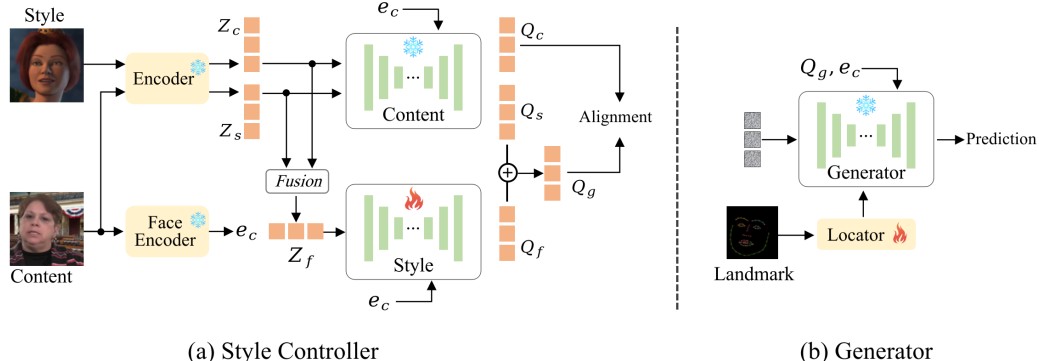

(a) Style Controller                                        (b) Generator

Figure 1: Framework of our proposed StyleFace. (a) Style controller (b) Generator.

## 3.1 FEATURE FUSION MODULE FOR STYLE-IDENTITY INTEGRATION

The prerequisite for a successful style transfer is the effective disentanglement of content and style information. We achieve this by modulating the attention mechanism within the diffusion U-Net, as

attention layers are instrumental in defining the spatial relationships and structural composition of the generated image.

Given a style image $I_s$ and an identity image $I_c$, the task is to generate an image that combines the style of $I_s$ with the identity information of $I_c$. Our process begins by extracting a high-fidelity identity embedding. We employ insightface [1] encoder $E_f$ to extract face identity embeddings $e_c = E_f(I_c)$, where $I_c$ denotes the content image. Throughout the entire style transfer process, we consistently use this face identity embedding as a conditional signal to maintain the identity information. Specifically, we utilize $e_c$ as the key and value features to apply cross-attention in all the attention layers of a pretrained diffusion U-Net.

For the style image and identity image, latent features are extracted by the encoder of a pretrained diffusion mdoel, denoted by $Z_s$ and $Z_c$, respectively. Both latent features are then processed via the U-Net with the identity embedding $e_c$ as the conditional signal, obtaining the features at each layers in the U-Net, denoted by $Q_s$ and $Q_c$, respectively. These features serve as the foundation for controlling stylization and identity perservation in the generating process.

Additionally, to mitigate the influence of style and content images from different domains, we further perform a fusion from a statistical properties perspective Huang & Belongie (2017),

$$Z_f = \sigma_s(Z_c - \mu_c)/\sigma_c + \mu_s, \tag{1}$$

where $\mu_c$, $\sigma_c$ and $\mu_s$, $\sigma_s$ denote the mean and standard deviation of the content latents $Z_c$ and style latents $Z_s$, respectively. The resulting fused features $Z_f$ are then processed through a trainable U-Net with the identity embedding $e_c$ as the conditional signal, obtaining the features at each layers, $Q_f$.

**Style-identity Feature Fusion.** While AdaIN modulates global style characteristics through statistical alignment, we observe that it primarily captures color distributions, leaving more nuanced stylistic elements unaddressed. To overcome this limitation, we propose a feature fusion mechanism that leverages the rich style information as well as the identity guideline encoded in $Q_s$ features Specifically, we introduce a controllable fusion between identity-preserving features $Q_f^i$ and style features $Q_s^i$:

$$Q_g^i = \alpha Q_f^i + (1 - \alpha)Q_s^i, \tag{2}$$

where $\alpha \in [0, 1]$ serves as a user-controllable parameter. This formulation enables precise control over the style-identity trade-off: when $\alpha$ approaches 1, the model preserves more identity features from the content image, while values closer to 0 result in stronger style transfer effects. Our experiments demonstrate that this simple yet effective approach achieves superior style transfer results while maintaining fine-grained control over identity preservation (see Section 4.6).

**Identity Preservation.** While the style controller effectively injects fused features $Q_g$ into the generator to guide the stylization process, the optimization of the LoRA modules still needs constriction to avoid identity degradation in the stylized outputs. To further enforce identity consistency, we introduce an identity alignment loss that constrains the query features of the LoRA module $Q_f^i$ to remain proximal to the corresponding query features of the content image $Q_c^i$:

$$\mathcal{L}_{\text{align}} = \frac{1}{N} \sum_{i=1}^{N} \|Q_f^i - Q_c^i\|_2, \tag{3}$$

where $N$ denotes the number of self-attention layers in the network architecture. This alignment loss effectively creates a regularization mechanism that preserves identity-critical features throughout the optimization process, mitigating identity distortion while still allowing for expressive style transfer.

## 3.2 STATISTICAL ALIGNMENT FOR IDENTITY AND STYLE PRESERVATION

Building on our disentangled representations, the core of our method is a targeted statistical transfer that operationalizes our central hypothesis: that the channel-wise statistics of deep features separately encode content and style. We posit that first-order statistics (the mean) capture low frequency, structural information that defines a subject's identity such as the overall facial structure and the arrangement of key features. In contrast, second-order statistics (the standard deviation) capture

---

[1] https://github.com/deepinsight/insightface

the high-frequency textural and chromatic variations that define an artistic style such as brushstroke patterns, color palettes, and other aesthetic details. This conceptual separation allows us to surgically manipulate the feature distributions, implanting a new style by modifying the standard deviation while preserving the subject's identity by leaving the mean intact.

We implement this principle via a statistical consistency loss applied to the predicted noise $\epsilon^t$ at each diffusion timestep $t$. By shaping the noise, we directly control the characteristics of the final generated image. The loss guides $\epsilon^t$ to simultaneously match the mean of the content image's latent noise $\epsilon_c^t$ and the standard deviation of the style image's latent noise $\epsilon_s^t$:

$$\mathcal{L}_{\text{style}} = \|\mu(\epsilon^t) - \mu(\epsilon_c^t)\|_2 + \|\sigma(\epsilon^t) - \sigma(\epsilon_s^t)\|_2, \tag{4}$$

where $\mu(\cdot)$ and $\sigma(\cdot)$ denote the channel-wise mean and standard deviation operations, respectively. The terms $\epsilon_c^t$ and $\epsilon_s^t$ represent the latent representations of the content and style images at time step $t$. This formulation ensures that the generated output maintains the content structure of the content image while adopting the stylistic characteristics of the reference style image.

The first term of the loss function constrains the mean statistics to match those of the content image, thereby preserving structural and identity features. Simultaneously, the second term enforces stylistic consistency by aligning the standard deviations with those of the style reference. This dual-objective optimization effectively disentangles content and style in the latent space during the diffusion process.

### 3.3 SEMANTIC-GUIDED ADVERSARIAL OPTIMIZATION

While the statistical transfer excels at aligning low-level stylistic properties like texture and color, it does not explicitly enforce global artistic coherence. A generated image might have the correct local statistics but fail to capture the overall composition, mood, or semantic structure of the target style. To bridge this gap, we introduce a semantic-aligned adversarial objective that guides the generative process toward holistic stylistic consistency. This objective leverages features from DINO Zhou et al. (2024), which excel at capturing high-level semantic information while preserving fine spatial details.

For the adversarial objective, pixel-based discriminators often fail at this task, as they tend to focus on superficial details rather than high-level artistic concepts. We therefore design a discriminator that operates not in pixel space, but within the rich semantic feature manifold of a Vision Transformer (ViT). By leveraging features from models pre-trained to understand image content and context, our discriminator assesses the stylization from a more human-aligned perspective, ensuring that the final output is not only statistically similar but also perceptually convincing.

We first estimate the denoised latent $\epsilon_0$ from the predicted noise $\epsilon_t$ at diffusion time step $t$ using the standard diffusion reverse process equation: $\epsilon_0 = \epsilon_t - \sqrt{1 - \alpha_t}\epsilon_t / \sqrt{\alpha_t}$, where $\epsilon^t$ is the predicted latent at $t$, and $\alpha_t$ represents the cumulative product of noise scheduling coefficients. The denoised latent $\epsilon_0$ is then decoded to image space as $x_0 = \mathcal{D}(\epsilon_0)$, where $\mathcal{D}$ denotes the latent decoder.

To enhance discrimination robustness and semantic awareness, we condition the discriminator on multi-level semantic features extracted from the generated image: CLIP embeddings $e_{\text{clip}}(x_0)$ capture high-level semantic concepts, while DINO features $f_{\text{dino}}(x_0)$ provide fine-grained visual information with strong spatial correspondence. The generator's adversarial objective is formulated as:

$$\mathcal{L}_{\text{adv}}^G = -\mathbb{E}_{x_0, t}[D_\theta(e_{\text{clip}}(x_0), f_{\text{dino}}(x_0))], \tag{5}$$

where $D_\theta$ is the discriminator parameterized by $\theta$.

The discriminator architecture consists of a feed-forward network that processes the features from CLIP and DINO alongside the pixel-level image representation. It is trained using a hinge loss formulation:

$$\mathcal{L}_{\text{adv}}^D = \mathbb{E}_{x_0, t}[\max(0, 1 - D_\theta(e_{\text{clip}}(x_0), f_{\text{dino}}(x_0)))] + \mathbb{E}_{x_s}[\max(0, 1 + D_\theta(e_{\text{clip}}(x_s), f_{\text{dino}}(x_s)))], \tag{6}$$

where $x_s$ represents reference style images. This adversarial mechanism encourages the model to generate outputs that are indistinguishable from authentic style references in both pixel space and semantic feature space, while the conditional design ensures that the stylization process remains semantically coherent with respect to both content and style references.

**Optimization:** The overall optimization objective is a weighted sum of the identity alignment loss $\mathcal{L}_{align}$, the statistical alignment loss $\mathcal{L}_{style}$ and the adversarial loss $\mathcal{L}_{adv}^D$:

$$\mathcal{L} = \lambda_1 \mathcal{L}_{align} + \lambda_2 \mathcal{L}_{style} + \lambda_3 \mathcal{L}_{adv}^D, \tag{7}$$

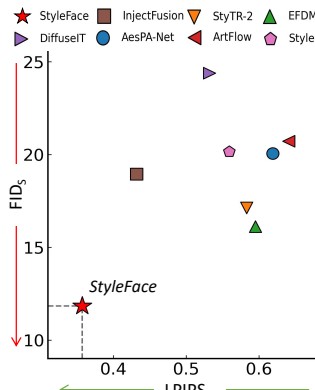

Figure 2: LPIPS *vs.* $\text{FID}_S$.

Table 1: Quantitative comparison between our StyleFace and seven state-of-the-art methods. **Bold** indicates best performance.

| Method | LPIPS ↓ | CLIP-I ↑ | $\text{FID}_S$ ↓ | $\text{FID}_C$ ↓ | ArtFID ↓ |
|---|---|---|---|---|---|
| AesPA-Net Hong et al. (2023) | 0.619 | 0.532 | 19.978 | 19.783 | 28.421 |
| EFDM Zhang et al. (2022a) | 0.595 | 0.526 | 16.119 | 19.051 | 24.326 |
| ArtFlow An et al. (2021) | 0.641 | 0.535 | 20.715 | 25.686 | 29.956 |
| StyTR$^2$ Deng et al. (2022) | 0.583 | 0.509 | 17.121 | **13.824** | 24.129 |
| DiffuseIT Kwon & Ye (2022b) | 0.532 | 0.653 | 24.390 | 21.378 | 34.941 |
| InjectFusion Jeong et al. (2024) | 0.432 | 0.670 | 18.946 | 26.186 | 31.179 |
| StyleID Chung et al. (2024) | 0.559 | 0.590 | 20.158 | 17.313 | 31.552 |
| **StyleFace** ($\alpha = 0.1$) | **0.357** | **0.882** | **11.838** | 27.185 | **21.481** |
| **StyleFace** ($\alpha = 0.2$) | 0.417 | 0.808 | 14.335 | 20.340 | 25.144 |
| | +17.4%↑ | +31.6%↑ | +26.6%↑ | – | +11.0%↑ |

where $\lambda_{1,2,3}$ are the weighted hyperparameters.

## 4 EXPERIMENT

### 4.1 EXPERIMENTAL SETTING

**Implement Details.** Our method is implemented using the Stable Diffusion architecture from the diffusers library [2] with a frozen pretrained generator and a trainable style controller. We integrate Low-Rank Adaptation (LoRA) modules with rank $r = 4$ into the U-Net blocks of the style controller, keeping the base weights fixed during optimization. Training is performed using the Adam optimizer with a learning rate of $1 \times 10^{-5}$ for both the style controller and discriminator networks, a batch size of 8, and for 1000 iterations on NVIDIA H800 GPUs. The style control strength parameter $\alpha$ is set to 0.2, and loss function weights are configured as $\lambda_1 = 2.0$, $\lambda_2 = 10.0$, and $\lambda_3 = 10.0$ based on validation performance. Complete details regarding the pretraining process for each components are available in Appendix B.1.

**Dateset.** The experiments use content images from the HDTF dataset Zhang et al. (2021) and style images from a cartoon dataset Pinkney & Adler (2020). We randomly select 25 frames from HDTF as content images and 20 samples from the cartoon dataset as style references. All faces are detected using the dlib face detector King (2009), aligned to the image center based on the detected facial landmarks, and then cropped or resized to 512×512 resolution. For quantitative evaluation, we generate 500 ($25 \times 20$) stylized images.

**Evaluation Metric.** We quantitatively evaluate the stylized images using four commonly used metrics: LPIPS Zhang et al. (2018) (Learned Perceptual Image Patch Similarity) to measure the perceptual similarity between stylized and style images, FID Heusel et al. (2017) (Fréchet Inception Distance) to access the overall quality of the generated images ($\text{FID}_S$ indicates the distance between stylized images and style images, $\text{FID}_C$ indicates the distance between stylized images and content images), CLIP-I Radford et al. (2021) (CLIP Image Similarity) that computes the latent-space alignment between stylized and style images to measure semantic-level consistency, and ArtFID Wright & Ommer (2022) for comprehensive assessment of coherence among stylized, content, and style images.

### 4.2 QUANTITATIVE EVALUATION

We quantitatively evaluate the proposed StyleFace and compare it with seven facial style transfer methods (the implement details of baseline methods can refer to Appendix B.2), including four conventional generative methods (AesPA-Net Hong et al. (2023), StyTR$^2$ Deng et al. (2022), EFDM Zhang et al. (2022a) and ArtFlow An et al. (2021)) and three diffusion-based methods (DiffuseIT Kwon & Ye (2022b), InjectFusion Jeong et al. (2024) and StyleID Chung et al. (2024)).

---

[2]https://github.com/huggingface/diffusers

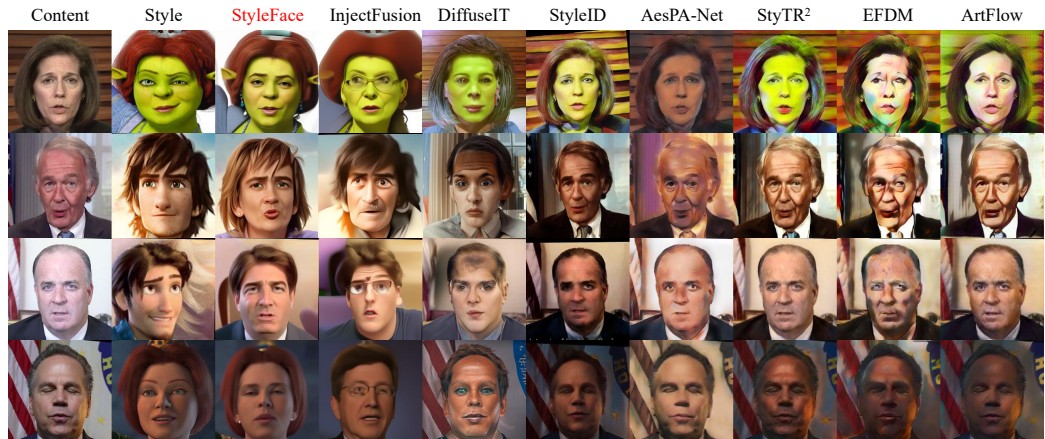

| Content | Style | StyleFace | InjectFusion | DiffuseIT | StyleID | AesPA-Net | StyTR$^2$ | EFDM | ArtFlow |

Figure 3: Qualitative comparison of facial stylization results.

Table 1 presents the quantitative results comparing our StyleFace with state-of-the-art methods. Our approach achieves the best performance on four of the five metrics, demonstrating its effectiveness in balancing identity preservation with style transfer. Specifically, StyleFace outperforms all competing methods on stylization performance (as presented in Figure 2), indicating superior style transfer fidelity of proposed StyleFace. For identity preservation, our method performs competitively, we achieve comparable results to GAN-based method.

We observe that conventional style transfer evaluation metrics have limitations when evaluating stylized facial images. The quantitative results in Table 1 don't fully align with the visualization results in Figure 3 (*i.e.*, the FID$_C$ metric compares content images with their stylized versions, inadequately measures identity preservation as it penalizes intentional artistic modifications). This occurs because effective facial stylization requires altering certain facial characteristics while maintaining identity that existing metrics cannot properly assess. To provide a more comprehensive evaluation that better reflects the performance of stylization, we further conduct a LLM-based evaluation to assess both identity preservation and style transfer quality in Section 4.4.

### 4.3 QUALITATIVE EVALUATION

**Visualization Results.** Figure 3 presents qualitative comparisons between our StyleFace and baseline methods (we present more comparison results in Appendix C.1). Our approach demonstrates superior performance in simultaneously preserving facial identity while effectively transferring artistic styles from reference images. Conventional generative methods (*i.e.* AesPA-Net, ArtFlow) often distort facial identity features when attempting to match reference styles. Diffusion-based approaches (*i.e.* DiffuseIT, StyleID) better preserve identity but exhibit limited stylistic diversity and expressiveness.

It's notable that there is a fundamental trade-off in existing methods: enhanced stylization typically comes at the cost of identity preservation. Our StyleFace effectively resolves this dilemma through identity preservation and statistical alignment to keep the identity and a adversarial learning to control the style transfer, achieving both superior style transfer fidelity (evidenced by better CLIP-I and FID$_S$ scores compared to diffusion-based methods) and remarkable identity preservation (demonstrated by improved ArtFID and FID$_C$ metrics relative to conventional approaches). This balanced performance confirms the effectiveness of our statistical alignment technique and adversarial optimization framework in navigating the challenging identity-style trade-off (more visualization results of our StyleFace can refer to Appendix C.2).

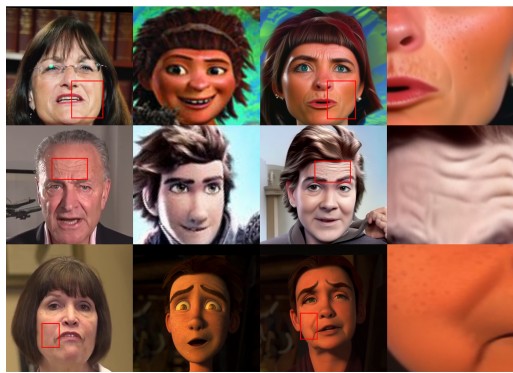

Figure 4: Detailed facial feature preservation.

**Face Detail Preservation.** We further present a detailed examination of facial feature preservation in Figure 4. Our StyleFace successfully maintains critical identity elements such as facial structure, distinctive features, and expressions while applying artistic stylization.

These results demonstrate the effectiveness of our proposed identity learning and statistical alignment techniques. Additionally, the high-quality stylization outcomes validate the contribution of our semantic-guided adversarial learning approach, which ensures coherent style transfer while preserving the subject's identity traits (more visual results can refer to Appendix C.2).

## 4.4 HUMAN EVALUATION

The face shape and color tone of content images are both changed in the stylized images. The existing metrics are insufficient for a comprehensive evaluation of the stylized images, particularly in capturing facial detail features and overall visual quality. Therefore, we conducted further human-involved evaluations to obtain a more thorough assessment. We mainly focus on three aspects: identity preservation (IP), style transfer quality (ST), and overall assessment (OA). We randomly selected 6 content-style pairs from our test set and generated 36 stylized images using our StyleFace and all the baseline methods (the details of the evaluation are shown in Appendix B.4).

Table 2: Human evaluation comparing our Style-Face with baseline methods across three criteria. All scores use a 5-point scale (1=worst, 5=best).

| Method | IP ($\uparrow$) | ST ($\uparrow$) | OA ($\uparrow$) |
|---|---|---|---|
| StyTR$^2$ | $4.39_{\pm 0.657}$ | $3.17_{\pm 0.745}$ | $3.50_{\pm 0.849}$ |
| EFDM | $3.54_{\pm 0.546}$ | $2.62_{\pm 0.916}$ | $2.62_{\pm 0.869}$ |
| StyleID | $4.45_{\pm 0.598}$ | $3.22_{\pm 0.583}$ | $3.44_{\pm 0.437}$ |
| StyleFace | $3.78_{\pm 0.629}$ | $\mathbf{4.72}_{\pm 0.416}$ | $\mathbf{3.67}_{\pm 0.408}$ |

Table 2 presents the results of the human-involved evaluations, showing that StyleFace consistently outperforms baseline methods. Specifically, it achieved an average overall assessment (OA) score of 3.67, much higher than baseline methods. Similarly, StyleFace demonstrates superior performance in style transfer quality compared to baseline methods and comparable identity preservation performance.

## 4.5 ABLATION STUDY

To evaluate the contribution of each component in our proposed StyleFace, we conduct a comprehensive ablation study as shown in Table 3 and Figure 5. Removing the statistical alignment component (w/o $\mathcal{L}_{align}$) results in a lower LPIPS score but slightly higher FID$_C$, indicating better stylization performance at the cost of reduced identity preservation. Without the adversarial learning component (w/o $\mathcal{L}_{adv}^{D,G}$), we observe significant degradation in style fidelity (CLIP-I drops from 0.808 to 0.777), confirming its role in maintaining overall stylization performance during style transfer. Similarly, removing the style learning objective (w/o $\mathcal{L}_{style}$) leads to comparable degradation patterns, with ArtFID increasing to 25.420 and CLIP-I decreasing to 0.775. The visual results in Figure 5 further illustrate these effects, with ablated variants showing either compromised identity preservation, inconsistent stylization, or unwanted facial distortions. These results collectively demonstrate that each component makes essential contributions to achieving the optimal balance between identity preservation and style transfer in our framework.

Content  Style  StyleFace  w/o $\mathcal{L}_{align}$  w/o $\mathcal{L}_{adv}^{D,G}$  w/o $\mathcal{L}_{style}$

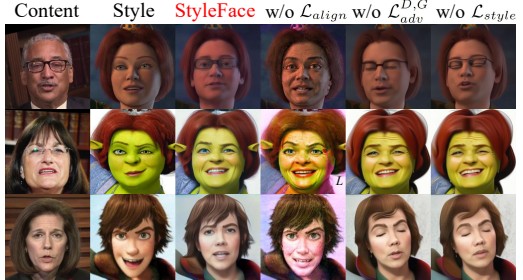

Figure 5: Visualization of ablation study.

Table 3: Ablation study on StyleFace components, evaluating the impact of removing statistical alignment ($\mathcal{L}_{align}$), adversarial learning ($\mathcal{L}_{adv}^{D,G}$), and style learning ($\mathcal{L}_{style}$).

| Objective | LPIPS $\downarrow$ | CLIP-I $\uparrow$ | FID$_S$ $\downarrow$ | FID$_C$ $\downarrow$ | ArtFID $\downarrow$ |
|---|---|---|---|---|---|
| w/o $\mathcal{L}_{align}$ | 0.320 | 0.840 | 13.966 | 22.630 | 25.127 |
| w/o $\mathcal{L}_{adv}^{D,G}$ | 0.447 | 0.777 | 14.372 | 20.060 | 25.288 |
| w/o $\mathcal{L}_{style}$ | 0.449 | 0.775 | 14.451 | 20.010 | 25.420 |
| StyleFace | 0.417 | 0.808 | 14.335 | 20.340 | 24.144 |

## 4.6 PARAMETER ANALYSIS

In this section, we analyze the performance of our method with different parameter settings (the effect of $\lambda_1$, $\lambda_2$ and $\lambda_3$ can refer to appendix B.5). We test the effect of the value of $\alpha \in [0, 1]$ (refer to Equation 2) on the stylization quality and identity preservation. We set $\alpha = \{0.1, 0.2, 0.4, 0.6, 0.8\}$) to evaluate the style transfer quality and identity preservation of our method on the test set. We plot the style fidelity score ($\text{FID}_S$) and identity preservation ($\text{FID}_C$) in Figure 6 (a), the visualization results are shown in Figure 6 (b). It can be seen a clear trade-off: increasing $\alpha$ improves identity preservation while reducing style transfer quality. Our experiments indicate that $\alpha = 0.2$ provides the optimal balance between these competing objectives.

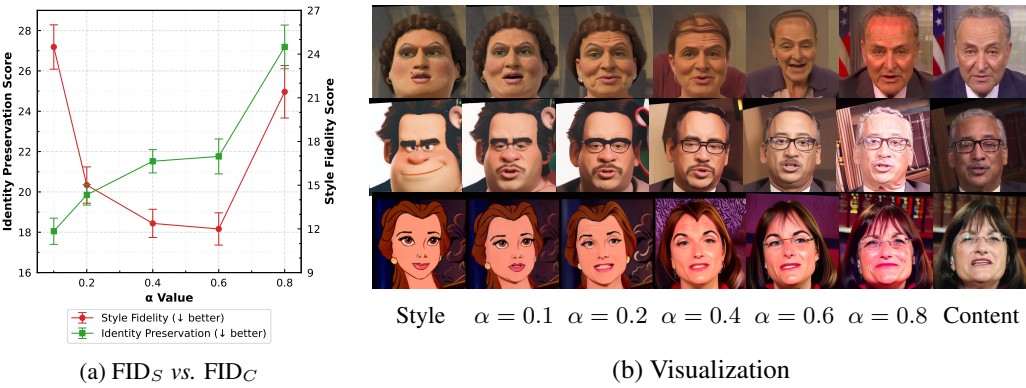

| (a) $\text{FID}_S$ *vs.* $\text{FID}_C$ | (b) Visualization |

Figure 6: The effect of parameter $\alpha$.

## 4.7 COMPUTATIONAL COST

We provide the computational cost of our StyleFace and the baseline methods in Table 4. We record the average inference time of our StyleFace and seven baseline methods when processing a single image on a single H800 GPU. The inference time is calculated by averaging the time required to generate 10 images (each generated one at a time, *i.e.*, with a batch size of 1). Our method achieves an inference time of only 1.699 seconds, which is significantly faster than other diffusion-based methods. Even compared to StyleID, a train-free diffusion method, our approach demonstrates better computational efficiency. This advantage stems from StyleID requiring both reverse and forward processes, while our method only needs a forward process. When compared to conventional style transfer methods, our approach is indeed slower, which is expected as diffusion-based methods generally require more computational resources than traditional style transfer techniques.

Table 4: The computational cost of our StyleFace and the baseline methods.

| Method | Ours | DiffuseIT | InjectFusion | StyleID | StyTR$^2$ | AesPA-Net | EFDM | ArtFlow |
|---|---|---|---|---|---|---|---|---|
| Infer. Time (s) | 1.699 | 226.823 | 68.243 | 3.971 | 0.200 | 0.469 | 0.002 | 0.516 |

## 5 CONCLUSION

In this paper, we presented StyleFace, a novel diffusion-based approach for face stylization that effectively balances identity preservation and style transfer. Our method addresses the fundamental challenge of simultaneously maintaining facial identity while achieving compelling artistic stylization. The key technical contributions of our approach include: an attention-based feature fusion module complemented by identity alignment mechanism that enable fine-grained control over identity-style balance through learnable modulation, a statistical alignment technique that leverages the complementary nature of channel-wise means and standard deviations to preserve content structure and stylistic attributes, and a semantic-based adversarial optimization to ensure semantic-level stylistic coherence. Our comprehensive experiments demonstrate that StyleFace consistently outperforms existing state-of-the-art methods in both identity preservation and style transfer.

## 6 ETHICS STATEMENT

Our work focuses on face stylization for artistic and creative applications. However, we acknowledge that, like any generative model technology, it could potentially be misused for creating synthetic media, also known as deepfakes. We condemn any malicious use of this technology. Our framework is not designed for photorealistic generation and produces stylized, non-realistic outputs, which we believe mitigates some of these risks. Furthermore, our model is trained and evaluated on publicly available datasets (e.g., HDTF), and we have not used any private data. We also recognize that the underlying pre-trained models (such as the base diffusion model, CLIP, and DINO) may inherit societal biases from their training data. While our method does not amplify these biases, performance may vary across different demographic groups. We encourage further research into fairness and bias in generative models.

## 7 REPRODUCIBILITY STATEMENT

To ensure the reproducibility of our results, we will make our source code and pre-trained model weights publicly available upon publication. Our framework is built upon publicly available models, including a Stable Diffusion v1.5 base model and pre-trained encoders such as insightface, CLIP, and DINO. All datasets used for training and evaluation are publicly available and will be specified in the experimental section.

The training procedure, including all hyperparameters such as the loss weights $(\lambda_1, \lambda_2, \lambda_3)$, learning rate, batch size, and optimizer settings, will be detailed in the appendix. We will also provide the specific self-attention layers that were modulated and the details of the LoRA implementation. The evaluation protocol, including the metrics used and the setup for user studies, will be described thoroughly to allow for fair and accurate comparisons.

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

**Use of LLMs.** We used a large language model (LLM) as a writing assistant to improve the clarity, conciseness, and overall quality of this manuscript. The LLM's role was primarily in rephrasing and refining the text to ensure a smooth, logical, and consistent narrative. The core technical contributions, experimental design, results, and analyses were conducted solely by the human authors. All content generated by the LLM was carefully reviewed, edited, and approved by the authors to ensure technical accuracy and alignment with our research.

**Boarder Impact.** Our face stylization StyleFace offers significant creative potential for digital artists, content creators, and the general public. By enabling high-quality artistic transformations while preserving identity, our technology can democratize creative expression, support the entertainment industry, and enhance personalized content creation across various platforms. We believe that our technology can be used for good, and we are committed to developing appropriate safeguards to minimize potential harms. We then discuss the potential negative impacts of our technology and propose several mitigation strategies.

**Negative Impacts.** We acknowledge several potential negative societal impacts directly related to our technology: (1) Disinformation the technology could be exploited to create misleading content or manipulated images that appear authentic but represent fictional scenarios; (2) Identity misrepresentation malicious actors might generate stylized fake profiles for deception or fraud; (3) The StyleFace could be applied to images without subjects' consent, potentially in ways that compromise dignity or autonomy; and (4) Dual-use concerns while designed for entertainment, the technology could potentially be repurposed for surveillance applications that identify individuals across different visual styles. We believe responsible research requires acknowledging these risks explicitly and implementing appropriate safeguards, as detailed in our mitigation strategies.

**Mitigation Strategies.** We propose several approaches to address these concerns: (1) implementing robust visible or invisible watermarking for all generated images to maintain provenance; (2) developing and releasing complementary detection tools that can identify images generated by our system; (3) establishing guidelines requiring explicit consent before processing personal images; (4) actively curating diverse training datasets representing various demographic groups and artistic traditions; (5) initially releasing our model through a gated API that monitors for potential misuse rather than releasing model weights directly. We are committed to ongoing evaluation of our technology's societal impact and will continue to refine our approach to maximize benefits while minimizing potential harms.

## A  INTEGRATION OF FACE EMBEDDING

As shown in Figure 7, we employ a pre-trained face encoder $E_f$ to extract content image identity embeddings $e_c = E_f(I_c)$, $I_c$ denotes the content image. This embedding serves as a conditioning signal that is strategically injected into the self-attention (SA) layers of the style controller. Specifically, we utilize $e_c$ as the key and value features for SA layers to preserve identity-specific characteristics. For style latent integration, we replace the query features of the SA layers with the content image's face embedding:

$$Attn(Q_s^i, e_c, e_c) = softmax(\frac{Q_s^i e_c^T}{\sqrt{d}})e_c, \quad (8)$$

where $d$ is the dimension of the query features $Q_s^i$.

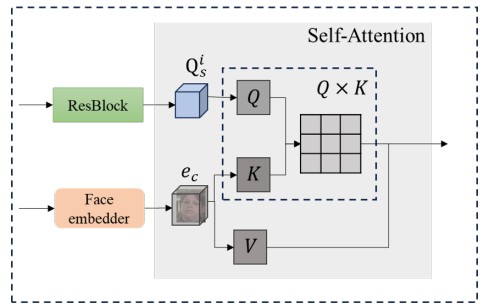

Figure 7: Architecture of the identity preservation module.

By incorporating the content image's face embedding into the self-attention layers, we ensure identity preservation throughout the network. As the self-attention operations progress from the downsampling to the upsampling layers, the identity information from the content image is gradually integrated into the style image's query features. This progressive integration allows for a balanced preservation of identity characteristics while still accommodating stylistic modifications, with identity information becoming increasingly prominent as the signal passes through successive self-attention layers.

We also use the same embedding for the content image to preserve the identity of the content image:

$$Attn(Q_c^i, e_c, e_c) = softmax(\frac{Q_c^i e_c^T}{\sqrt{d}})e_c, \tag{9}$$

where $Q_c^i$ is the query features of the SA layers of the content image.

## B EXPERIMENT

### B.1 DETAILS OF EXPERIMENT SETTING

Our StyleFace is implemented using the Stable Diffusion architecture from the diffusers library [3] with a frozen pretrained generator and a trainable style controller. We integrate Low-Rank Adaptation (LoRA) modules with rank $r = 4$ into the U-Net blocks of the style controller, keeping the base weights fixed during optimization. The LoRA modules are initialized with the original pretrained weights of the SA layers. And the other components of the style controller are still keep frozen with the original pretrained weights. The code of our proposed StyleFace is presented in https://anonymous.4open.science/r/style_transfer_2-7D43.

**Training of LoRA modules:** Training is performed using the Adam optimizer with a learning rate of $1 \times 10^{-5}$, $\beta_1 = 0.9$, $\beta_2 = 0.999$, weight decay of $1 \times 10^{-2}$, and epsilon of $1 \times 10^{-8}$. We use a constant learning rate scheduler with gradient accumulation steps of 1, and a maximum gradient norm of 1.0. We use mixed precision with weight_dtype of "fp16", while xformers memory-efficient attention is enabled. The training runs for 1000 steps without gradient checkpointing. For the scheduler, we use DDIM scheduler with uncond_ratio to 0.1, noise_offset to 0.05, snr_gamma to 5.0, and enable_zero_snr to True. The style control parameter $\alpha$ is set to 0.2, and loss function weights are configured as $\lambda_1 = 2.0$, $\lambda_2 = 10.0$, and $\lambda_3 = 10.0$ based on validation performance.

**Training of Discriminator:** The discriminator is trained using the Adam optimizer with same hyper-parameters as the training of LoRA modules. The discriminator is trained at the first 300 steps and the generator begins training after the first 300 steps, with an adversarial loss weight of 10.0 for the generator. We employ a hinge loss function and we don't use the penalty for the discriminator. For feature extraction, we utilize DINO features from layers 2, 5, 8, and 11 to enhance the discriminator's perceptual capabilities.

**Pretraining Details:** We pretrain the style controller and generator together as a unified system to establish effective control relationships between them. For our implementation, we leverage the pretrained face recovery model from Xu et al. (2024b), which provides both the style controller and generator components. As illustrated in Figure 8, our pipeline operates with a clear division of responsibilities: the style controller processes only the face image input, while the generator takes random noise as its primary input. The generator then conditions its output on two key elements: (1) the feature representations extracted by the style controller and (2) the face embedding. This architecture enables the system to generate faithful reconstructions of the original face images.

### B.2 THE IMPLEMENTATION OF BASELINE METHODS

All the baseline methods are implemented based on the official repositories. And all the methods are evaluated on the same datasets: content images selected from the HDTF dataset Zhang et al. (2021) and style images selected from the cartoon dataset Pinkney & Adler (2020). The detailed settings are as followed.

**AesPA-Net Hong et al. (2023).** We followed the official repository[4] and evaluated it under the default configuration.

**EFDM Zhang et al. (2022b).** We followed the sub-section '*ArbitraryStyleTransfer*' in official repository[5] and evaluated it under the following configuration: crop enabled, preserving color enabled and $\alpha = 1.0$.

---

[3]https://github.com/huggingface/diffusers

[4]https://github.com/Kibeom-Hong/AesPA-Net

[5]https://github.com/YBZh/EFDM

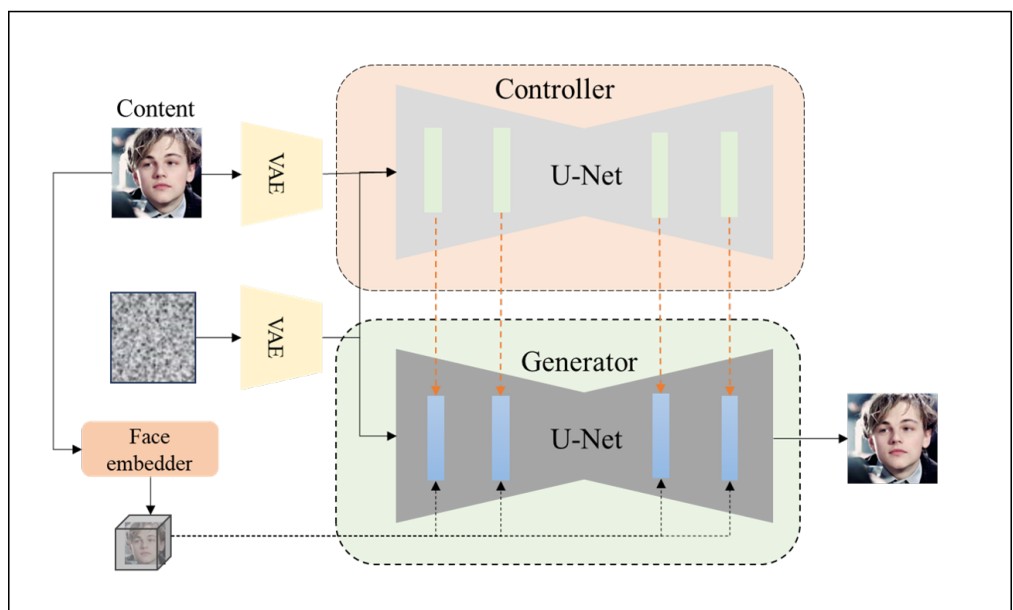

Figure 8: The pipeline of the style controller and the generator.

**ArtFlow An et al. (2021).** We followed the official repository[6] and evaluated it under the following configuration: crop = true, using adain as the operator, number of flows in each block (8) and number of blocks (2).

**StyTR$^2$ Deng et al. (2022).** We followed the official repository[7] and evaluated it under the following configuration: position embeddings is set to sine, size of the embeddings (512), and $\alpha = 1.0$.

**DiffuseIT Kwon & Ye (2022b).** We followed the official repository[8] and evaluated it under the following configuration: range restart enabled, noise augmentation enabled, color match enabled, 100 diffusion iterations, 200 timestep respacing and 80 skip timesteps.

**InjectFusion Jeong et al. (2024).** We followed the official repository [9] and evaluated it under the following configuration: 1000 generation steps, 50 inversion steps, lambda coefficient for sampling calibration (0.9985), Slerp ratio (0.3), style calibration (0.0), quality boosting (200) and mask enabled.

**StyleID Chung et al. (2024).** We followed the official repository[10] and evaluated it under the following configuration: query preservation hyperparameter (5), and attention temperature scaling hyperparameter (1.5).

### B.3 DETAILS OF HUMAN EVALUATION

We conducted human-involved evaluations to assess the overall quality of the stylized images from three different perspectives. (identity preservation, style transfer ability, and overall assessment), comparing our StyleFace with previous methods. The template we used for evaluation is shown in Figure 10. We randomly select 6 content images and 6 style images, and then generate 36 stylized images per method for evaluation. Each image was assessed by two different volunteers, with a total of 14 volunteers participating in the evaluation. The experimental results for human evaluation are provided in Table 6

---

[6]https://github.com/pkuanjie/ArtFlow
[7]https://github.com/diyiiyiii/StyTR-2
[8]https://github.com/cyclomon/DiffuseIT
[9]https://github.com/curryjung/InjectFusion_official
[10]https://github.com/jiwoogit/StyleID

Table 5: Human evaluation comparing our StyleFace with baseline methods across three criteria. All scores use a 5-point scale (1=worst, 5=best).

| Method | IP ($\uparrow$) | ST ($\uparrow$) | OA ($\uparrow$) |
|---|---|---|---|
| AesPA-Net | $4.46_{\pm 0.519}$ | $2.04_{\pm 0.691}$ | $2.75_{\pm 0.661}$ |
| EFDM | $3.54_{\pm 0.546}$ | $2.62_{\pm 0.916}$ | $2.62_{\pm 0.869}$ |
| ArtFlow | $4.46_{\pm 0.431}$ | $2.88_{\pm 0.740}$ | $3.17_{\pm 0.773}$ |
| StyTR$^2$ | $4.39_{\pm 0.657}$ | $3.17_{\pm 0.745}$ | $3.50_{\pm 0.849}$ |
| DiffuseIT | $2.92_{\pm 0.812}$ | $3.38_{\pm 0.703}$ | $2.79_{\pm 0.828}$ |
| InjectFusion | $2.04_{\pm 0.749}$ | $4.04_{\pm 0.557}$ | $2.67_{\pm 0.624}$ |
| StyleID | $4.45_{\pm 0.598}$ | $3.22_{\pm 0.583}$ | $3.44_{\pm 0.437}$ |
| StyleFace | $3.78_{\pm 0.629}$ | $\mathbf{4.72}_{\pm 0.416}$ | $\mathbf{3.67}_{\pm 0.408}$ |

Figure 9: The template used for human-involved evaluation.

## B.4    LLM-BASED EVALUATION

We leverage the powerful image understanding capabilities of large models to evaluate the generated images across three perspectives (identity preservation, style transfer ability, and overall assessment), comparing our StyleFace with previous methods. The LLM we used in ChatGPT-4o, and the prompt we used for evaluation is shown in Figure 10. We randomly select 10 content images and 10 style images, and then generate 100 stylized images per method for evaluation. The experimental results for LLM-based evaluation are provided in Table 6

## B.5    PARAMETER ANALYSIS

**The effect of** $\lambda_1$    Figure 11 demonstrates that our StyleFace maintains consistent performance across different values of $\lambda_1$. The evaluation metrics exhibit minimal variation throughout the tested range, indicating the robustness of our approach to this hyperparameter. Our selected value achieves superior results across multiple quality indicators, including the lowest LPIPS, highest $\text{CLIP}_I$ score, lowest $\text{FID}_S$, and lowest ArtFID among the examined configurations. While this optimal setting results in a slightly elevated $\text{FID}_C$ compared to alternatives, the substantial improvements in other metrics justify this minor trade-off. These findings suggest that our chosen $\lambda_1$ configuration successfully establishes an ideal balance between effective style transfer and faithful identity preservation within our framework.

Table 6: LLM-based evaluation comparing our StyleFace with baseline methods across three criteria. All scores use a 5-point scale (1=worst, 5=best).

| Method | IP ($\uparrow$) | ST ($\uparrow$) | OA ($\uparrow$) |
|---|---|---|---|
| AesPA-Net | $3.75_{\pm 0.433}$ | $3.68_{\pm 0.464}$ | $3.50_{\pm 0.500}$ |
| EFDM | $3.81_{\pm 0.386}$ | $3.63_{\pm 0.481}$ | $2.70_{\pm 0.900}$ |
| ArtFlow | $3.85_{\pm 0.448}$ | $3.67_{\pm 0.718}$ | $3.48_{\pm 0.500}$ |
| StyTR$^2$ | $4.05_{\pm 0.394}$ | $3.89_{\pm 0.718}$ | $3.64_{\pm 0.479}$ |
| DiffuseIT | $3.23_{\pm 0.697}$ | $3.85_{\pm 0.361}$ | $3.23_{\pm 0.421}$ |
| InjectFusion | $2.83_{\pm 0.460}$ | $3.93_{\pm 0.365}$ | $3.17_{\pm 0.378}$ |
| StyleID | $4.00_{\pm 0.343}$ | $3.64_{\pm 0.478}$ | $3.47_{\pm 0.587}$ |
| StyleFace | $3.63_{\pm 0.484}$ | $\mathbf{4.10}_{\pm 0.294}$ | $\mathbf{4.00}_{\pm 0.392}$ |

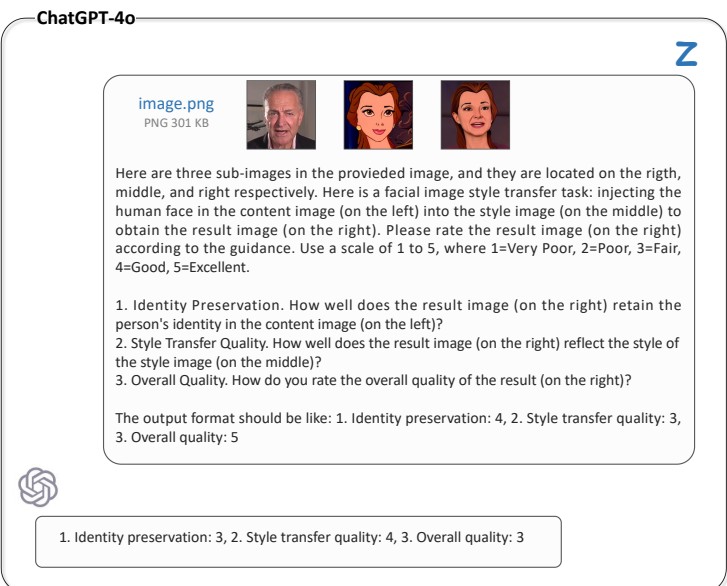

Figure 10: The template used for LLM-based evaluation.

**The effect of $\lambda_2$.** Figure 12 illustrates that our StyleFace exhibits robustness to variations in $\lambda_2$, with performance metrics remaining largely stable across different values. Quantitative results show only minor fluctuations, indicating low sensitivity to this hyperparameter. Our selected value of $\lambda_2 = 10$ achieves optimal performance across multiple metrics, yielding the lowest LPIPS, highest CLIP$_I$ score, lowest FID$_S$, and lowest ArtFID among tested values. While this setting produces a marginally higher FID$_C$ compared to alternatives, the improvement in other metrics suggests that $\lambda_2 = 10$ effectively balances style transfer quality with identity preservation. The comparable performance of alternative values ($\lambda_2 \in \{1, 5, 15, 20\}$) further confirms the method's stability, with $\lambda_2 = 10$ providing the most favorable trade-off for our stylization objectives.

**The effect of $\lambda_3$** The effect of $\lambda_3$ is shown in Figure 13. We can see that the performance of our StyleFace is not sensitive to the choice of $\lambda_3$. The metrics remain relatively stable across different values, with only minor fluctuations. Notably, our chosen value of $\lambda_3 = 10$ achieves the best performance with the lowest LPIPS, highest CLIP$_I$ score, lowest FID$_S$, and lowest ArtFID, although it has a slightly higher FID$_C$ compared to other values. The other $\lambda_3$ values (1, 5, 15, and 20) show very similar performance to each other. This suggests that $\lambda_3 = 10$ provides the optimal balance between style transfer quality and identity preservation in our method.

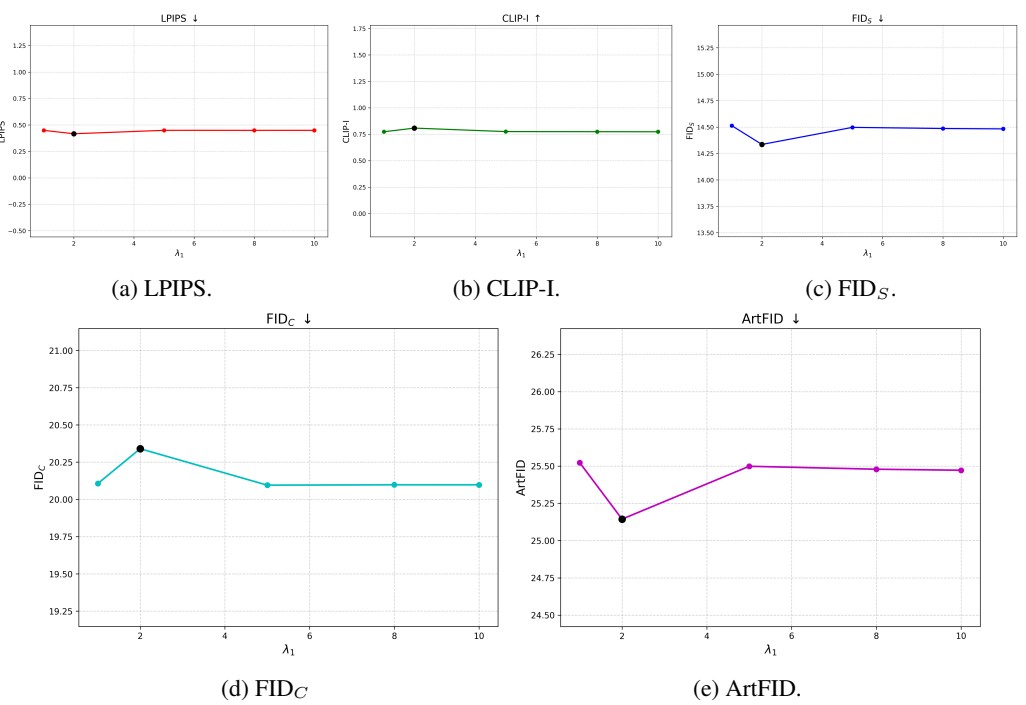

(a) LPIPS.  (b) CLIP-I.  (c) FID$_S$.

(d) FID$_C$  (e) ArtFID.

Figure 11: The effect of $\lambda_1$.

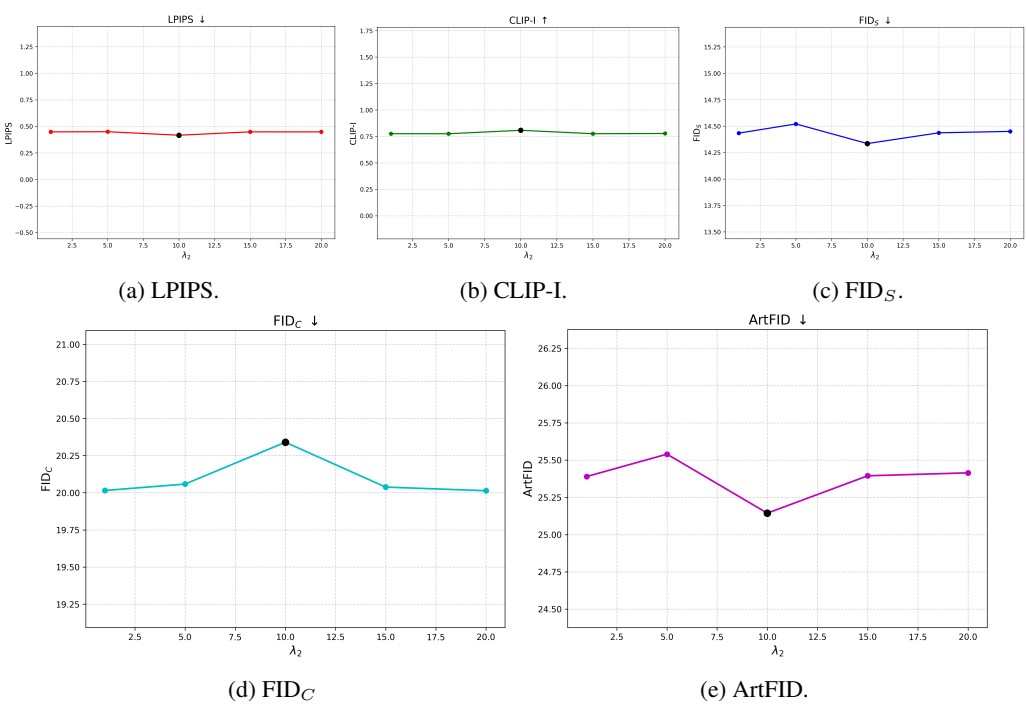

(a) LPIPS.  (b) CLIP-I.  (c) FID$_S$.

(d) FID$_C$  (e) ArtFID.

Figure 12: The effect of $\lambda_2$.

## C  VISUALIZATION

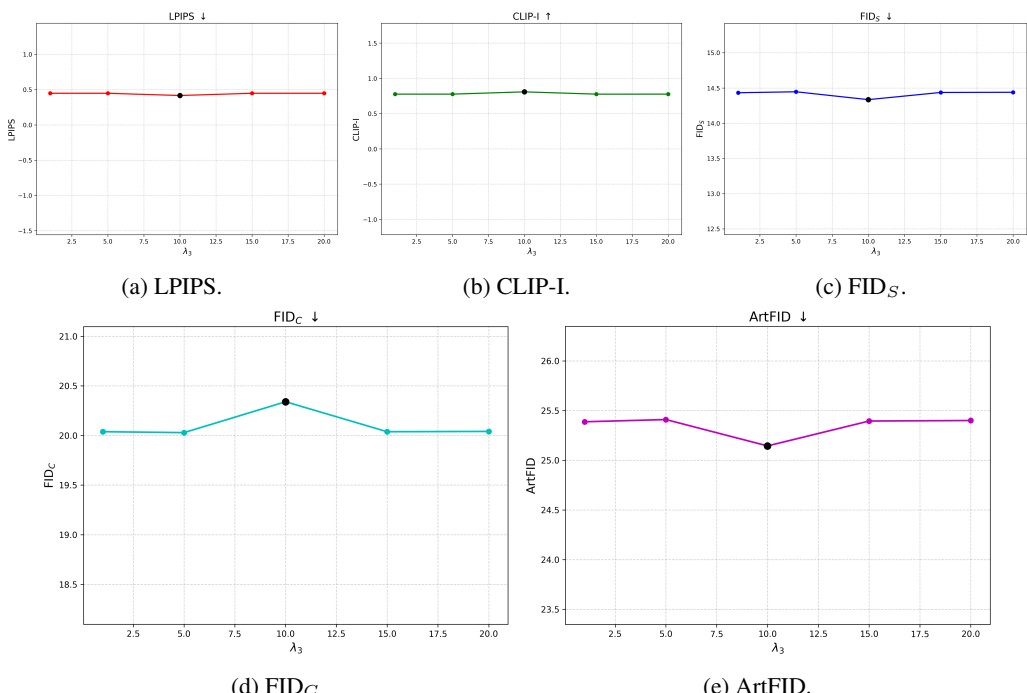

Figure 13: The effect of $\lambda_3$.

## C.1 MORE COMPARISON RESULTS

Figure 14 shows more visualizations of the images stylized by our StyleFace. And in Figure 15, we present more cases to visually compare our StyleFace with baseline methods. These examples demonstrate our method's consistent style transfer performance and its effectiveness in simultaneously preserving the identity features of the content image while capturing the artistic characteristics of the style image.

## C.2 MORE VISUALIZATION

**More face detail preservation visual results:** As shown in Figure 16, we provide more visual results to show the facial details preservation of our StyleFace. From Figure 16 (a) to (d), the visual results of our StyleFace are shown as teeth, facial wrinkles, eye shape, and mouth shape, respectively. The red boxes highlight the details that our method preserves. It demonstrates that our StyleFace can preserve the key facial details of the content image while still maintaining the style of the style image, which is beneficial for the preservation of the identity of the content image. **More style type visualization:** We provide more style type visualization to show the versatility of our StyleFace. We use images from MetFace Karras et al. (2020a) dataset as the style images. MetFace dataset contains art images from different styles, including cartoon, watercolor, oil painting, and sketch. In Figure 17, we show the stylization results of our StyleFace on different art style images. It can be seen that our StyleFace can preserve the key facial details of the content image while still maintaining the style of the style image on the art style images, which demonstrates the generalization of our StyleFace.

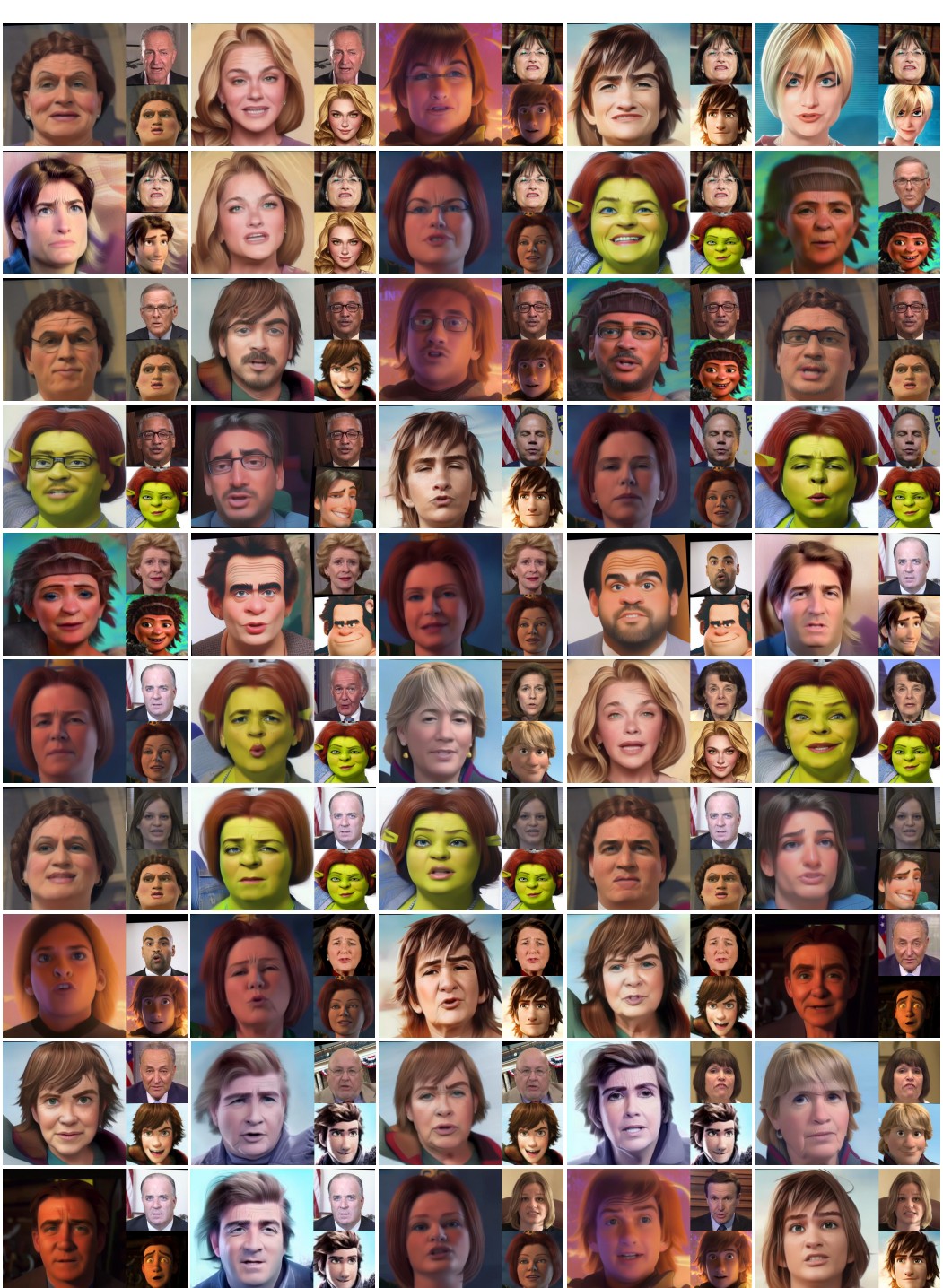

Figure 14: Qualitative evaluation of StyleFace

Content   Style   StyleFace   InjectFusion   DiffuseIT   StyleID   AesPA-Net   StyTR²   EFDM   ArtFlow

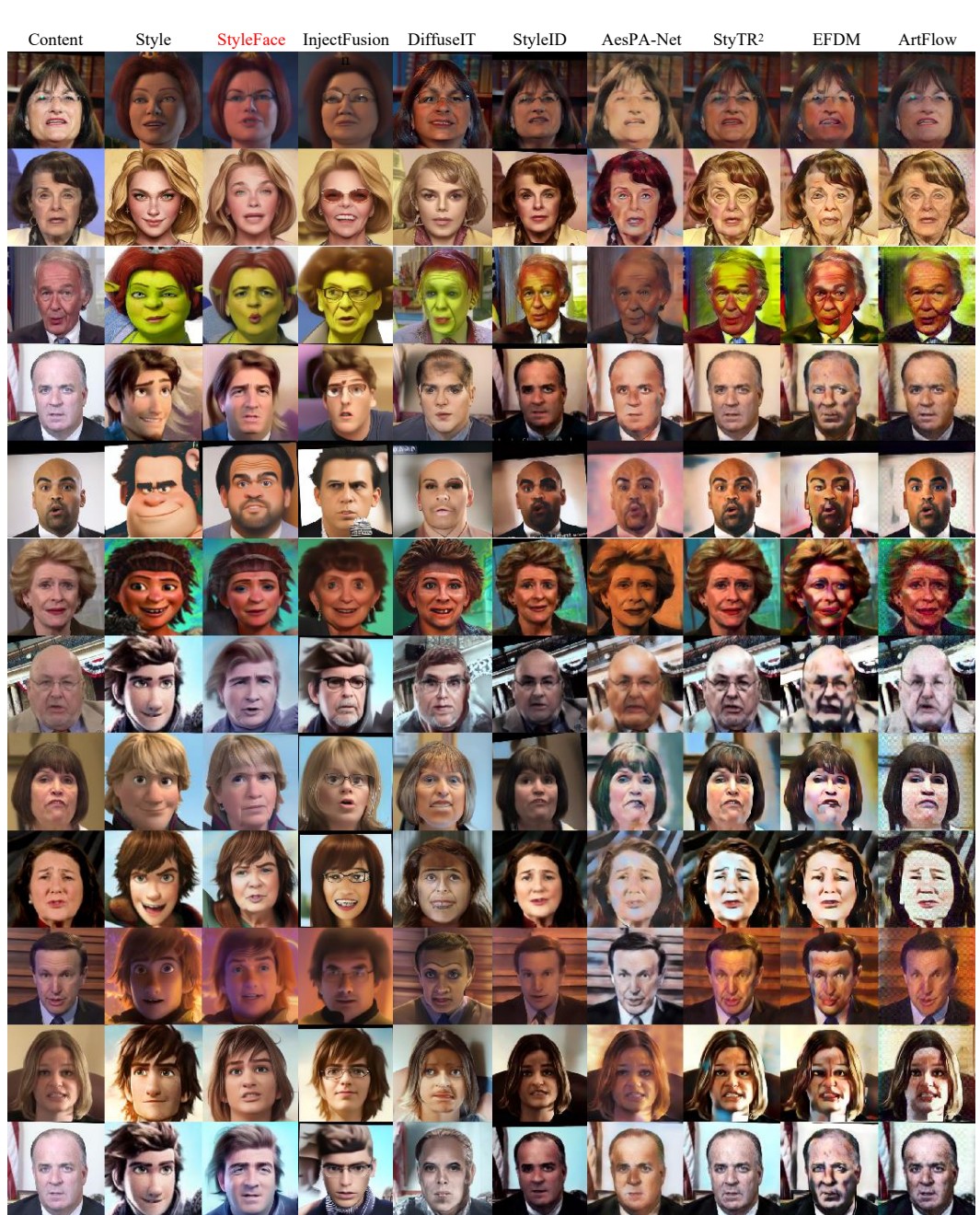

Figure 15: Visualization: comparing StyleFace with baseline methods.

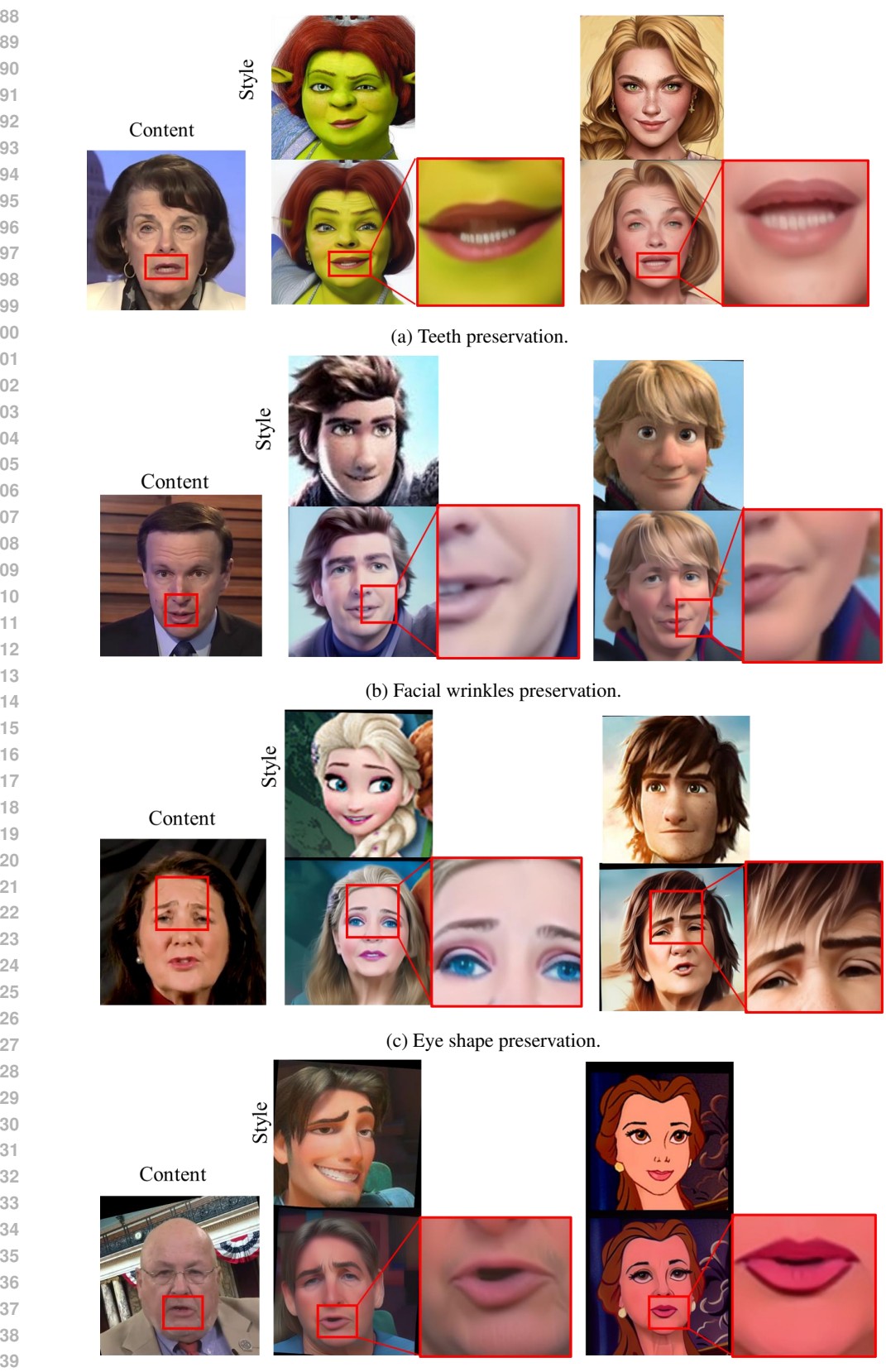

(a) Teeth preservation.

(b) Facial wrinkles preservation.

(c) Eye shape preservation.

(d) Mouth shape preservation.

Figure 16: The facial details preservation of our StyleFace. From Figure 16 (a) to (d), the visual results of our StyleFace are shown as teeth, facial wrinkles, eye shape, and mouth shape, respectively. The red boxes highlight the details that our StyleFace preserves.

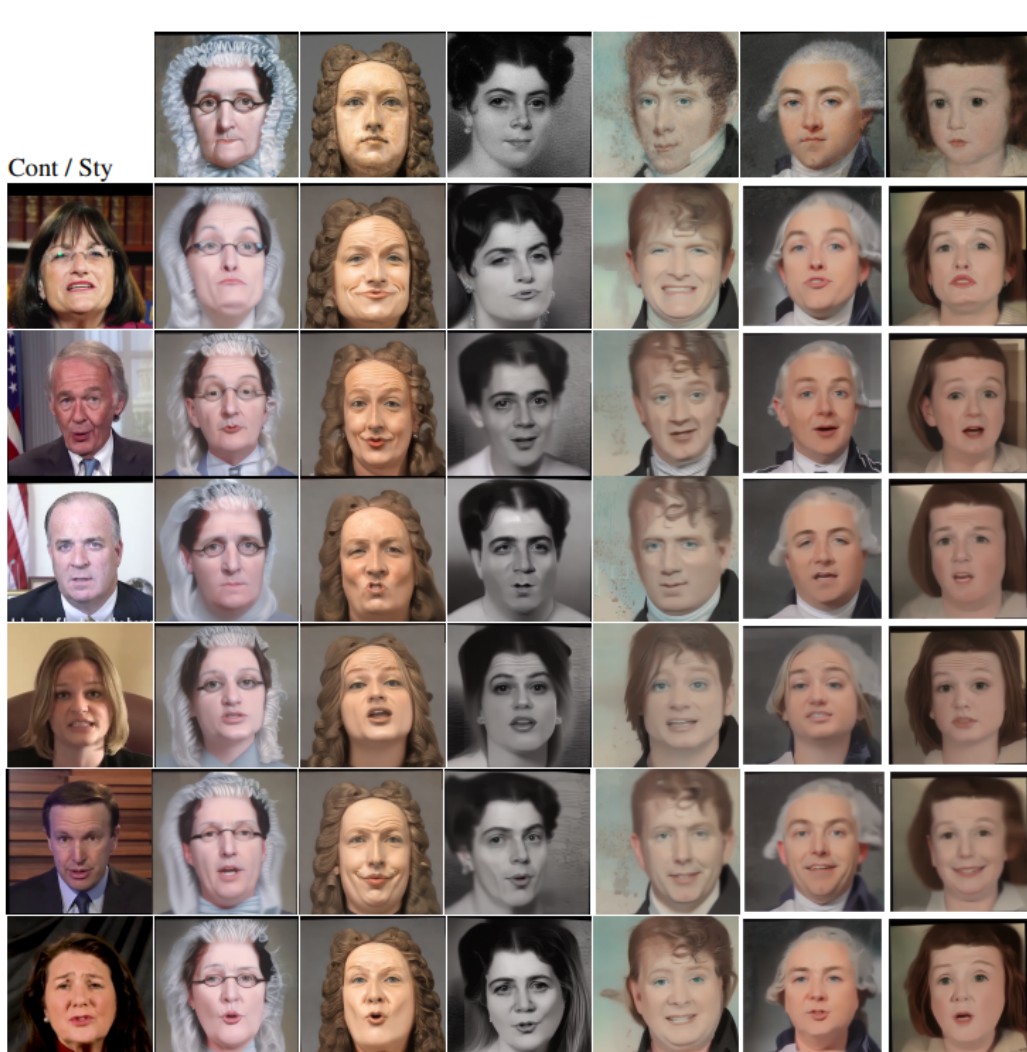

Figure 17: The stylization results of different art style images.

