# OpenReview forum: "Forging a Masterpiece from Any Face: A Universal Framework for Face Stylization"
_ICLR.cc/2026/Conference — ICLR 2026 Conference Withdrawn Submission_

### Official Review · Reviewer_frMZ · 2025-10-26

**Soundness:** 3
**Presentation:** 3
**Contribution:** 2
**Rating:** 4
**Confidence:** 5

**Summary:**

This paper introduces StyleFace, a novel diffusion-based framework for face stylization that aims to disentangle content (identity) from style to achieve high-quality stylized outputs while preserving identity. The core contributions include a disentangled attention module, a statistical style injection layer, and a perceptually-aligned adversarial objective. The authors claim that StyleFace outperforms existing methods in terms of balancing identity preservation and style fidelity based on both quantitative metrics and human evaluation.

**Strengths:**

1. The paper clearly articulates the fundamental trade-off in face stylization between style fidelity and content preservation, a well-recognized challenge in the field.
2.  The authors present extensive quantitative comparisons using a suite of metrics (LPIPS, CLIP-I, FIDs, ArtFID) and conduct a human evaluation, which is valuable for subjective tasks like stylization. The ablation study is also a positive aspect, attempting to justify the contribution of each component.
3. The inclusion of a detailed reproducibility statement and the intention to release code and pre-trained models is highly commendable and beneficial for the research community.

**Weaknesses:**

1. While the framework combines several existing techniques (diffusion models, attention mechanisms, statistical alignment, adversarial objectives), the specific combination and the individual components do not appear to offer sufficient novelty. Many aspects feel like a re-application or slight modification of established ideas in the context of face stylization.
2. Despite claims of "unprecedented balance" and "consistently outperforms," the provided qualitative results in Figure 3 and Figure 15 are not overwhelmingly superior to baselines, and in some cases, the differences are subtle or even debatable. The "Stylized Face" example on page 1 already raises concerns about the strength of style transfer. The human evaluation scores, while higher for StyleFace in ST and OA, show a lower IP score compared to some baselines (e.g., StyTR2, StyleID, AesPA-Net), which contradicts the core claim of balancing identity preservation. Based on the qualitative results provided, it is difficult to distinguish the stylized facial identity information.
3. The descriptions of the "disentangled attention module" and "statistical style injection layer" lack the depth needed to fully understand their unique contributions beyond generic concepts. For example, how does the attention module orthogonally project information, and what specific statistical properties are manipulated beyond just mean and standard deviation in a novel way for face stylization?
4. The Figure 1 is very confusing, as it appears to be a fusion of the embeddings generated by the face encoder and the embeddings generated by the diffusion model encoder.

**Questions:**

1. You mention a "statistical style injection layer that manipulates feature distributions." How does this layer differ significantly from existing adaptive instance normalization (AdaIN) or similar statistical alignment methods? What specific "feature distributions" are manipulated, and how does this manipulation uniquely preserve identity while implanting style, beyond simply aligning mean and standard deviation?
2. The human evaluation results in Table 2 show that StyleFace has a lower Identity Preservation (IP) score compared to several baseline methods (StyTR2, StyleID, AesPA-Net). Given that "identity preservation" is a core claim and a key strength of your framework, how do you reconcile these lower scores with your assertion of "unprecedented balance between identity preservation and style fidelity"? Could you provide more detailed analysis or alternative interpretations of these specific results?

---

### Official Review · Reviewer_n7J2 · 2025-10-30

**Soundness:** 2
**Presentation:** 2
**Contribution:** 1
**Rating:** 2
**Confidence:** 5

**Summary:**

This paper presents a targeted statistical transfer method for arbitrary face stylization. In particular, the authors use channel-wise mean of deep features to maintain identity and deep features’ standard deviation for style reference. In addition, the whole framework consists of a disentangled attention module and a statistical style injection layer. The entire process is optimized using a perceptually-aligned adversarial objective.

**Strengths:**

(1) The motivation is well presented of using the proposed targeted statistical transfer for arbitrary face stylization.

(2) The explanations and illustrations are mostly clear and intuitive of the disentangled attention, the statistical style injection layer, the perceptually-aligned adversarial objective.

**Weaknesses:**

(1) The novelty and contribution are very limited. In particular, the generator’s framework is largely a copy of the IP-Adapter where cross-attention is employed to inject FaceID features. The essence of style controller is the same with AdaIN where mean and standard variance are used for feature alignment.

(2) As for the experimental results, the claimed identity-style disentanglement is not achieved according to the face stylization results.

(3) On Page 1, Line 030-045, the figure has no legend and not cited in the main text. There is no explanation or analysis on the figure, and the face stylization is poor due to unexpected FaceID inconsistency between the content image and the stylized image, not to mention facial attributes’ inconsistency.

(4) On Page 4, Line 167, the authors employed insightface for FaceID feature extraction. This is not included in the abstract and the authors seem to omit this important factor for ID preservation while the so-called statistical transfer are more pronounced.

(5) On Figure 6, the effect of parameter alpha shows inconsistent transition and lacks necessary analysis.

(6) Lack of related works. In literature, there has been a lot of GAN and Diffusion-based methods for arbitrary face stylization e.g. BlendGAN, JoJoGAN, IP-Adapter, InstantID. These works are neglected for comparison.

[1] Liu, Mingcong, Qiang Li, Zekui Qin, Guoxin Zhang, Pengfei Wan, and Wen Zheng. "BlendGAN: Implicitly gan blending for arbitrary stylized face generation." Advances in neural information processing systems 34 (2021): 29710-29722.

[2] Chong, Min Jin, and David Forsyth. "JojoGAN: One shot face stylization." In European Conference on Computer Vision, pp. 128-152. Cham: Springer Nature Switzerland, 2022.

[3] Ye, Hu, Jun Zhang, Sibo Liu, Xiao Han, and Wei Yang. "IP-Adapter: Text compatible image prompt adapter for text-to-image diffusion models." arXiv preprint arXiv:2308.06721 (2023).

[4] Wang, Qixun, Xu Bai, Haofan Wang, Zekui Qin, Anthony Chen, Huaxia Li, Xu Tang, and Yao Hu. "InstantID: Zero-shot identity-preserving generation in seconds." arXiv preprint arXiv:2401.07519 (2024).

**Questions:**

No.

---

### Official Review · Reviewer_4RXM · 2025-10-31

**Soundness:** 3
**Presentation:** 3
**Contribution:** 2
**Rating:** 2
**Confidence:** 4

**Summary:**

This paper proposes a balanced face stylization framework for identity preservation and style transfer. Upon the powerful prior of the diffusion architecture, the proposed disentangling strategy and statistical manner reinforced each properties into generated face, outperforming existing SoTAs.

**Strengths:**

- Visual quality: The proposed method upon diffusion models demonstrates improved visual performance, outperforming previous models. The generated samples are seamless and relatively no visual artifacts.
- Better quantitative results: StyleFace showed significantly improved metrics, particularly on $CLIP-I$ and $FID_s$.
- Computing time: Unlike previous models, the proposed system takes substantial benefit in inference times, reporting about 2 second on a single H800 GPU, even though it is based on denoising diffusion scheme. Such advantages bring enhanced practicality for wide-ranging applications.

**Weaknesses:**

- Concerns on structure preservation: In experimental results, it is observed several stylized faces have different structures against content images. In specific, the left-bottom sample in first page and second row in Fig.3 demonstrated substantially changed head posture. This issues include not only head posture but also eye gazing, eye closing, glasses, expression and tooth (Fig.4). Such structure discrepancies raise significant concerns about structural fidelity. It is strongly recommended to discuss aforementioned fidelity concerns and evaluate the proposed system with more explicit metric to validate the structural preservation capabilities such as head-pose, facial expression or landmark distance.

- Missing comparison: The main comparison figures include image style transfer models rather than face stylization models. Thus, several face-central approaches are not included in experiments, e.g., JoJoGAN, DualStyleGAN, UI2I-Style, etc. Although the authors discussed these approaches in Sec.2.2, the proposed system was just compared with image-based style injection models. Considering the domain of this paper is in face, it is valuable to compare it to face stylization-centered models. Moreover, it lacks on literature analysis about this task even though face stylization has been relatively long-standing problem in generative realm, for instance, Toonify, Cross-Domain Style Mixing, CartoonGAN, AgileGAN. It seems that most related works are focused on diffusion-based approaches. The authors are recommended to compare it to face-focused models and discuss above issues. Also, it is crucial to analyze comprehensive studies in related works for potential readers to comprehend historical developments in this field.


(Miscellaneous)
- Pipeline: The proposed framework consists of several steps with diverse loss terms. For intuition, it is recommended to add algorithm to describe entire process or numerate the process step-by-step.
- Discussion about limitation: It is crucial to discuss some limitations of the proposed system, e.g., failure cases, bottle neck or further improvements.
- Inconsistent term: The authors interchangeably use the notation. For example, in Sec.3.1. $I_c$ is notated as ‘content image’ but simultaneously as ‘identity image’ as well. Also, Eq.(4) and (7) uses different font style to denote $L_{style}$. It is better to make consistent expressions to prevent the potential confusion.
- Typo and error: Removing unnecessary factors in manuscript is important. For instance, Fig.(5) includes unidentified alphabet in the 2nd row and 4th column as $L$. Also, there is no best-score notation in Tab.(2) for $IP$ score, degrading consistency.

**Questions:**

(Recommendation)
- It is better to provide additional results of face-centered stylization models for experiments and add some discussions about it including various StyleGAN-approaches.
- Considering significant concerns about structural fidelity mentioned above, it is imperative to alleviate this issue. If there is no strong preservation about input image, the fundamental goal of this task might be diluted.
- It is also important to analyze the comprehensive literatures in related works, especially on face stylization task.

---

### Official Review · Reviewer_Z6Ai · 2025-10-31

**Soundness:** 3
**Presentation:** 3
**Contribution:** 3
**Rating:** 6
**Confidence:** 2

**Summary:**

This paper introduces StyleFace, a new framework that treats face stylization as a targeted statistical transfer within a disentangled feature space. Experiments demonstrating that the proposed model outperforms state-of-the-art methods

**Strengths:**

- This manuscript is well-written, easy to understand, and easy to follow.

- This method is computationally effective, with an inference time of 1.699s on a single H800 GPU (Table 4), StyleFace is significantly faster than other diffusion-based methods, enhancing practical applicability.

- The performance of this article is excellent, especially in the experiments of Fig.2 and Table 1, which showed huge advantages.

**Weaknesses:**

- The main observation of this article: first-order statistics (i.e., channel-wise mean) of deep features primarily encode an image’s structural and identity information. This is a commonly used conclusion and is very common in early Style transfer work, such as classic work AdaIN. This method is based on AdaIN, which seems to be an incremental contribution.

- This method relies on meta information, such as the semantics of different style domains, which may not be available in practice.

- Reliance on Stable Diffusion and LoRA limits exploration of applicability to other generative architectures, restricting generalizability beyond these frameworks.

- The trade-off parameter α requires manual adjustment (Section 4.6), with optimal α=0.2 determined empirically, lacking an adaptive mechanism for user-friendly control.

- The experiments are not extensive enough and are only conducted on the Art dataset. If they can add more diverse tasks, such as face aging, human to animals, or add more art domains in existing experiments, such as sketch, it may be more solid.

**Questions:**

See weakness

---

### Note · Authors · 2025-11-13

**Comment:**

We thank all reviewers for their valuable efforts and insightful suggestions. We have carefully considered each comment and will take the useful comments into account to improve our work.

**Withdrawal Confirmation:**

I have read and agree with the venue's withdrawal policy on behalf of myself and my co-authors.